# Examining the Prevailing Negative Sentiments Related to COVID-19 Vaccination: Unsupervised Deep Learning of Twitter Posts over a 16 Month Period

**DOI:** 10.3390/vaccines10091457

**Published:** 2022-09-02

**Authors:** Qin Xiang Ng, Shu Rong Lim, Chun En Yau, Tau Ming Liew

**Affiliations:** 1Health Services Research Unit, Singapore General Hospital, Singapore 169608, Singapore; 2NUS Yong Loo Lin School of Medicine, National University of Singapore, Singapore 117597, Singapore; 3Department of Psychiatry, Singapore General Hospital, Singapore 169608, Singapore; 4SingHealth Duke-NUS Medicine Academic Clinical Programme, Duke-NUS Medical School, Singapore 169857, Singapore; 5Saw Swee Hock School of Public Health, National University of Singapore, Singapore 117549, Singapore

**Keywords:** machine learning, sentiment analysis, negative sentiment, COVID-19, vaccines

## Abstract

Despite the demonstrated efficacy, safety, and availability of COVID-19 vaccines, efforts in global mass vaccination have been met with widespread scepticism and vaccine hesitancy or refusal. Understanding the reasons for the public’s negative opinions towards COVID-19 vaccination using Twitter may help make new headways in improving vaccine uptake. This study, therefore, examined the prevailing negative sentiments towards COVID-19 vaccination via the analysis of public twitter posts over a 16 month period. Original tweets (in English) from 1 April 2021 to 1 August 2022 were extracted. A bidirectional encoder representations from transformers (BERT)-based model was applied, and only negative sentiments tweets were selected. Topic modelling was used, followed by manual thematic analysis performed iteratively by the study investigators, with independent reviews of the topic labels and themes. A total of 4,448,314 tweets were analysed. The analysis generated six topics and three themes related to the prevailing negative sentiments towards COVID-19 vaccination. The themes could be broadly understood as either emotional reactions to perceived invidious policies or safety and effectiveness concerns related to the COVID-19 vaccines. The themes uncovered in the present infodemiology study fit well into the increasing vaccination model, and they highlight important public conversations to be had and potential avenues for future policy intervention and campaign efforts.

## 1. Introduction

In our continued fight against the Coronavirus Disease 2019 (COVID-19) and the highly transmissible betacoronavirus, COVID-19 vaccines have been an important tool in protecting the vulnerable populations and reducing deaths and severe illness due to COVID-19 [1]. Despite the durable immunity offered by COVID-19 vaccines and reassuring public safety data, numerous studies have highlighted a significant amount of vaccine hesitancy among the general population, as well as overwhelmingly negative sentiments towards COVID-19 vaccination [2,3].

Today, social media including Twitter enjoy a high penetration rate and large numbers of daily active users [4], and the COVID-19 pandemic appears to have further fuelled its use for publicising viewpoints regarding vaccination effectiveness and safety [3]. Unfortunately, previous studies found rampant and persistently negative sentiments and misinformation on Twitter, which may adversely influence individual views and result in vaccine hesitancy or refusal [2,4,5]. Globally, vaccination rates have stagnated in some countries and, overall, the vaccination coverage still falls short of the initial target of 70% set by the World Health Organisation (WHO) in 2021 [6].

As highlight by an earlier study [2], which analysed temporal variations in the prevalence of the barriers and facilitators to COVID-19 vaccination over an 11 week study period, the barriers for individuals appeared to be consistent over time. This necessitates further investigation into the prevailing negative sentiments surrounding COVID-19 vaccination. Understanding the reasons for the public’s negative opinions towards COVID-19 vaccination using Twitter may help make new headways in improving vaccine uptake. Previous studies found such social media analyses to be a feasible and novel method to study public sentiment and emotional manifestations on a given topic [2,7,8]. In the case of COVID-19 vaccines, the conversations may be greatly polarising and incite intense feelings of varying opinions. Nonetheless, gaining insights into vaccination-hesitant subpopulations could aid future policy directions and intervention efforts.

This study, thus, aimed to examine the key negative sentiments towards COVID-19 vaccination that prevailed in the community via the analysis of public twitter posts over a 16 month period (1 April 2021 to 1 August 2022). As COVID-19 vaccine acceptance rates are known to change over time [9], the current investigation would provide the most recent update on the current status of vaccine hesitancy, globally. In doing so, policymakers can become more attuned to the public’s negative sentiments and work to address these concerns.

## 2. Methods

### 2.1. Study Design and Sample

In this infodemiology study, original tweets posted in English language from 1 April 2021 to 1 August 2022 were extracted. Retweets and duplicate tweets (i.e., tweets with identical sentence and words) were excluded from study. There was no restriction in the country of origin of the tweets, as long as the tweets were posted in English language.

### 2.2. Natural Language Processing and Thematic Analysis

Bidirectional encoder representations from transformers (BERT), a state-of-the-art deep machine learning approach for natural language processing (NLP), uses unsupervised masked language model (MLM) and the unsupervised next sentence prediction (NSP) for text deep pretraining and fine-tuning [10] compared to the traditional bag-of-words model in NLP. For MLM, a percentage of words are masked at random, and then these masked words are predicted on the basis of their bidirectional context, whereas the NSP model pretrains the text to understand relationships between sentences [10]. Unlike other text analysis approaches that require text to be pre-processed as part of data preparation prior to the analysis, the BERT-based model does not require much text pre-processing as the sentence context is provided by BERT.

Next, the named entity recognition, which recognises location, organisations, person, and miscellaneous entities, was used to select individual users only [11]. Individual Twitter users were identified by the use of actual human names on the Twitter account of each post. The SieBERT, a pretrained sentiment in English analysis model, was applied, and only negative sentiments tweets were selected [12]. Topic modelling, specifically BERTopic [13], was employed to generate coherent key concerns on the public discourse on COVID-19 vaccination. The data processing and machine learning approach is summarised in Figure 1.

Lastly, thematic analysis was performed iteratively with independent reviews of the topic labels and themes [14]. Coding disagreements were resolved through discussion amongst the study authors until a consensus was reached.

## 3. Results

A total of 25,232,314 initial tweets were identified in the period of 1 April 2021 to 1 August 2022. After removing duplicate tweets, tweets by organisations, and tweets without relevant terms of “COVID-19”, “vaccine”, or “vaccination”, a final 4,448,314 tweets remained. The flowchart showing the tweet selection process is illustrated in Figure 2.

The current analysis generated six topics related to the prevailing negative public sentiments towards COVID-19 vaccination. The total prevalence rate of these six topics was 93.3%; the majority of the tweets were centred around Topic 1, containing criticisms regarding vaccination passports. The topics were grouped into three main themes by manual thematic analysis. Table 1 contained the details of the topics within each theme. The remaining 6.7% of tweets were from a topic that was omitted from the current results as the BERT NLP model generated a miscellaneous topic that grouped all remaining (unfitted) tweets together.

We also analysed the temporal trend for these themes, as a function of the number of tweets posted for each topic over the 16 month study period (Figure 3). Theme 1 consisted of the combined tweets posted under Topics 1 and 2, Theme 2 consisted of the combined tweets posted under Topics 3 and 4, and Theme 3 consisted of the combined tweets posted under Topics 5 and 6. These trends were relatively constant over time. Predictably, tweets pertaining to concerns about the effectiveness of the COVID-19 vaccines against emerging variants appeared to coincide with the surge in Delta (July to August 2021) and Omicron (December 2021) cases in the United States (US).

## 4. Discussion

In this infodemiology study, unsupervised machine learning was utilised to analyse a large volume of free-text data from social media tweets, and the arising broad themes were further categorised through iterative thematic analysis. The prevailing negative sentiments towards COVID-19 vaccines could be broadly understood as either emotional reactions or safety and effectiveness concerns related to the vaccines.

The three main themes outlined in Table 1 were consistent with previous studies on COVID-19 vaccine hesitancy [2,3] and conceptual frameworks concerning individuals’ attitudes, enablers, and barriers to COVID-19 vaccination [15,16]. Using the increasing vaccination model (IVM), which has been frequently used by the WHO, US Centres for Disease Control and Prevention (CDC), academics, and practitioners, we can see how individual thought processes, social processes, and practical issues affect one’s willingness to be vaccinated [17]. The themes uncovered in the present study fit well into the IVM (Figure 4), and they highlight potential avenues for future policy intervention and campaign efforts. 

As illustrated in Figure 4, Themes 2 and 3 relate to an individual’s risk appraisal and confidence in the COVID-19 vaccine, and these thoughts and feelings motivate one to get vaccinated or not. Theme 2 highlights the public concerns regarding the risk of post-vaccination myopericarditis, which tends to happen in young males [18]. In the design of public communications surrounding the COVID-19 vaccines, it may help to acknowledge these concerns and explain that post-vaccination myopericarditis cases are rare and may happen with non-COVID-19 vaccines as well (with comparably low incidence rates) [18]. It is important to contextualise these risks to background rates and communicate it effectively to the public as public risk perception is particularly disposed to scare stories and fearmongering. A no-fault vaccine injury compensation programme may also help provide further reassurance to the public [19].

In addition, Theme 2 shows that vaccine misinformation and conspiracy theories on social media are widespread and would affect confidence in current vaccines and fuel vaccine hesitancy. It is important to remember that vaccination for even childhood diseases has always been an emotionally charged issue in many communities, even prior to the development of the COVID-19 vaccines [20]. For such topics, which has also become highly politicised, people may tend to be convinced by anecdotes and their own personal experience, which they consider unimpugnable as opposed to mainstream data and statistics [21]. This could also be related to the perceived censorship of any opposing views and lack of public discourse on vaccines, which may in fact weaken public confidence in the science and safety of COVID-19 vaccines and lead people to an echo chamber of “anti-vaxxer” views. A 2022 qualitative study of 26 well-established researchers, practicing doctors, and trained nurses reported perceived suppression of dissent in the field of COVID-19 vaccines, and the respondents raised concerns about a potential “backfire effect”, i.e., a counter-reaction to the lack of debate and contrary data in the public space that draws more attention to the “anti-vaxxer” position [22]. As such, members of the public should be provided with an open platform to ask questions and have their concerns addressed.

Theme 3 reflects the public concerns’ regarding the new variants of concern (VOCs) and the potential lack of efficacy of the ancestral COVID-19 vaccines against these new VOCs. As it is certainly possible that the SARS-CoV-2 virus could continue to evolve greater immune evasion and reduced vaccine effectiveness, it is helpful to have an honest and open communication with the public. Overconfident communication could result in public mistrust if expectations regarding disease prevention are not met. For example, it may be useful for authorities to plainly explain that neutralisation titres against the omicron variant are lower than earlier variants, and these lower titres could lead to an increased risk of severe breakthrough infection; however, a booster dose would mitigate this [23]. The focus should be on severe COVID-19 illness or intensive care unit (ICU) occupancy metrics rather than the absolute number of COVID-19 cases. Due to the novelty of the virus and the fact that there remain several unknowns and uncertainties surrounding SARS-CoV-2, the feelings of hesitancy and apprehension regarding VOCs and vaccine effectiveness are understandable, and it would be important for authorities and healthcare providers to actively listen and clarify these concerns.

Theme 1 reflects the “social outrage” in response to public policies and interventions aimed at increasing vaccine uptake. A high percentage of the tweets identified in this study were related to the introduction of vaccine passports. Some countries have either introduced or are contemplating mandatory COVID-19 certification or passports. Although such policies could directly promote behaviour change and increase vaccine uptake [24], they may also create invidious comparisons (especially with implementation and equity issues): “I think it’s more to do with the millions upon millions of people who’ve had covid and recovered. Basically, this just means there’s zero justification for vaccine mandates and passports (unless you’ve got shares in big pharma of course).” They may also produce strong negative sentiments as a result of perceived disparity and curtailment of individual freedom. Vaccine mandates or their equivalents are powerful tools to bolster vaccination uptake; however, governments need to consider the negative reactions and think through the implementation of such policies, weighing pre-existing COVID-19 vaccination coverage and hesitancy, as well as the pandemic trajectory. There is also evidence that such coercive COVID-19 vaccine policies may further erode public trust and vaccine confidence and exacerbate socioeconomic inequalities [25]. These policies may also alienate those who have yet to receive the vaccine and make them even less likely to do so [26].

In the same vein, the perceived unfair treatment of unvaccinated athletes, who are viewed to be the epitome of good health, could affect the public sentiments towards COVID-19 vaccines. This ranges from policies of requiring athletes to get vaccinated before they are allowed to train or participate in competitions: “So everyone is vaccinated but when someone gets COVID they will still have to quarantine and miss games.” This also concerns the perceived notion that the vaccines are unnecessary for athletes as they do not prevent COVID-19 transmission and are only applicable for persons with severe comorbidities: “OK #medtwitter, I need a little help. I’m seeing athletes (and others) who have been vaccinated against COVID, and are now testing positive (sometimes multiple times) on screening tests, and then being banned from events, jobs, etc. What gives? Is transmission a real concern?” There are also athletes who have been vocal about their refusal to get vaccinated; an example is Josh Archibald, a professional National Hockey League player who declined vaccination and subsequently became infected by COVID-19 and suffered COVID-related myocarditis [27]. Although this may motivate some people to get vaccinated, it may also stir negative feelings towards perceived “vaccine mandates” as unvaccinated players were sidelined indefinitely for not complying with local and state regulations.

Nevertheless, the findings of the present study should be interpreted in light of some limitations. First, the analysis was based solely on Twitter posts (with the majority of users hailing from North America and Europe), and only tweets in English were eligible for inclusion; hence, the findings should not and cannot be generalised to all populations and communities. Moreover, public sentiments may also differ significantly depending on local socio-political factors and the pandemic situation at that timepoint. Second, an unsupervised machine learning approach and Python package were employed to process the social media data and identify tweets with “probably” negative sentiments. This was based on probability, and the unsupervised machine learning approach cannot detect sarcasm; hence, misclassification can occur as a result, although the methods have good supporting evidence for their utility and face validity [2,28]. Third, nonhuman Twitter users (i.e., bots) masquerading as legitimate users and with the intention to distort public opinions [29] could have been included in our study sample. Despite best effort attempts to include tweets by users with actual human names and excluding retweets and duplicate tweets, a small number of these nonhuman tweets could have been included in our final study sample and unduly influenced the topic modelling process.

## 5. Conclusions

In conclusion, this study examined the prevailing negative public sentiments towards COVID-19 vaccines and highlighted important conversations with critical public health and policy implications. These sentiments appear to stem mainly from emotional reactions to perceived invidious policies or safety and effectiveness scepticism related to the vaccines. For future work, topic-based sentiment analysis of texts from other social media platforms including Facebook and Reddit can also be studied. Other methods can also be compared, as it is possible that newer techniques of sentiment analysis may yield greater accuracy.

## Figures and Tables

**Figure 1 vaccines-10-01457-f001:**
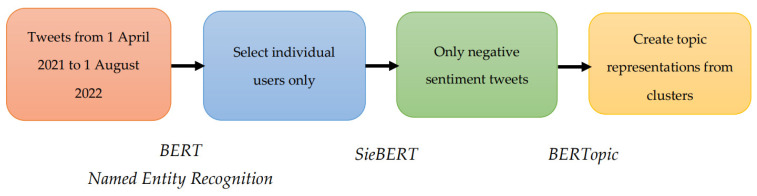
Unsupervised machine learning of free-text data from Twitter using BERT.

**Figure 2 vaccines-10-01457-f002:**
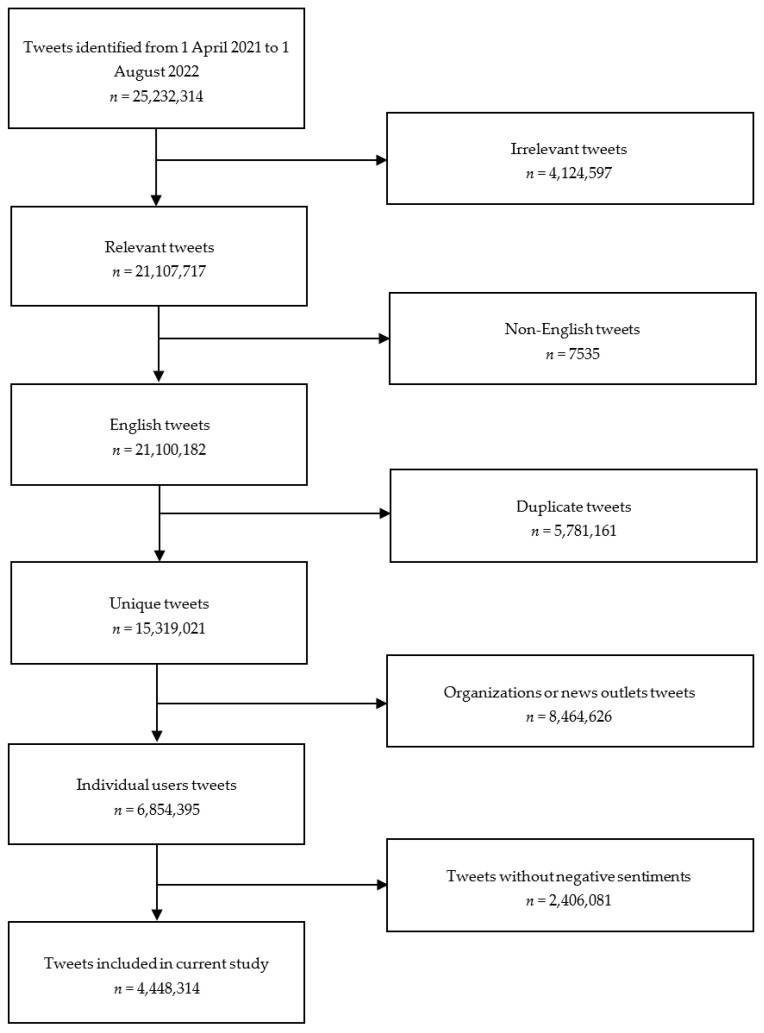
Flowchart illustrating tweet selection process.

**Figure 3 vaccines-10-01457-f003:**
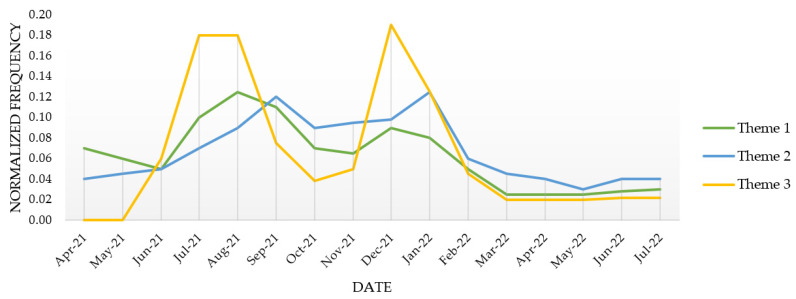
Temporal variations in the normalised frequency of tweets belonging to Theme 1 (Topics 1 and 2), Theme 2 (Topics 3 and 4), and Theme 3 (Topics 5 and 6).

**Figure 4 vaccines-10-01457-f004:**
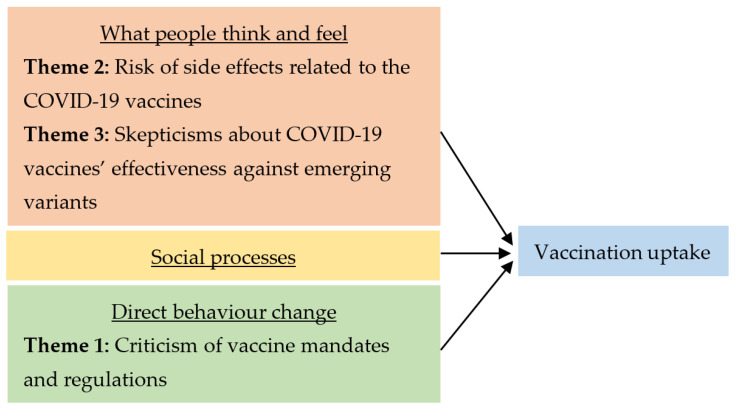
An illustration of how the themes fit into the increasing vaccination model [17].

**Table 1 vaccines-10-01457-t001:** Themes related to the negative public perception of COVID-19 vaccination, along with the respective topics and sample tweets (*n* = 4,448,314).

**Theme And Topic (*Keywords*)**	**Sample Tweets**	**Number of Tweets, *n* (%)**
**Theme 1: Criticism of vaccine mandates and regulations**		
Topic 1: Criticisms on the introduction of COVID-19 vaccine “passports” (*mask*, *Biden*, *Trump*, *wear*, *variants*, *home*, *COVID deaths*, *Americans*, *experimental*, *FDA*)	“Were it for a lethal virus I might understand your viewpoint but covid has a very low IFR and most who catch it only have mild symptoms. Mandating/coercing people to take the vaccine is unwarranted and unethical. Vax passports, a form of coercion, even more so.”	3,939,735 (88.6)
Topic 2: Citing unfair treatment of unvaccinated athletes (*players*, *NFL*, *athletes*, *team*, *game*, *player*, *games*, *vaccinated players*, *football*, *NBA*)	“This is wrong for so many for moral reasons. I hope the players refuse to play, you shouldn’t be threatened over a vaccine. It’s their choice. Unvaccinated players may face $14,650 fine from the league every time they violate COVID-19 procedures.”	61,556 (1.4)
**Theme 2: Risk of side-effects related to the COVID-19 vaccines**		
Topic 3: Misinformation regarding vaccine side-effects (*woman*, *mask*, *Trump*, *stupid*, *thinks*, *vaxxer*, *nurse*, *anti-vaxxer*, *job*, *positive COVID*)	“My relatives who were vaccinated died when they got COVID! The ones who got the antibodies lived when they got COVID! She is full of shit along with the FDA and Fauci!”	54,447 (1.2)
Topic 4: Risk of heart inflammation induced by the COVID-19 vaccines (*myocarditis*, *inflammation*, *heart inflammation*, *pericarditis*, *myocarditis COVID*, *risk myocarditis*, *rare*, *myocarditis vaccine*, *cardiac*, *myocarditis pericarditis*)	“The risk of myocarditis from COVID is a lot greater than the risk of myocarditis from the vaccine.”	37,944 (0.9)
**Theme 3: Scepticism about COVID-19 vaccines’ effectiveness against emerging variants**		
Topic 5: Concerns about the COVID-19 delta strain (*COVID delta*, *delta variant*, *COVID delta variant*, *strain*, *variants*, *19 delta*, *vaccinated delta*, *COVID 19 delta*, *delta COVID*, *alpha*)	“The delta variant of the Coronavirus is 7 times more deadly than the coronavirus, the only real protection any of us have is to get vaccinated, please don’t be stupid get vaccinated!”	31,895 (0.7)
Topic 6: Scepticism about COVID-19 vaccine effectiveness against the omicron strain (*omicron variant*, *variants*, *omicron COVID*, *COVID omicron*, *omnicron*, *omicron variant*, *new variant*, *protection omicron*, *getting omicron*, *omicron mild*)	“Given that blood clotting has been identified as a possible side-effect of COVID vaccines. I’d really start looking at them rather than blaming Omicron....”	21,883 (0.5)

## Data Availability

The datasets generated during and/or analysed during the current study are available from the corresponding author on reasonable request.

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
