# Peer review of "Examining the Prevailing Negative Sentiments Related to COVID-19 Vaccination: Unsupervised Deep Learning of Twitter Posts over a 16 Month Period"

_vaccines, 2022, doi:10.3390/vaccines10091457_

Round 1

Reviewer 1 Report

The authors use tools available in the literature (e.g.,  SieBERT) to classify sentiment in tweets.

(A) I'm unsure about what is the main paper's novelty. Is it only about the classification of the three themes?

(B) Themes 2 and 3 are insignificant statistically compared to theme 1. It means that almost all posts are in Theme 1, and your classification tool is worthless because, in almost all cases, the correct topic is Theme 1.

(C) Look at other studies about sentiment analysis related to Covid-19, and improve your review on the topic; an example is http://dx.doi.org/10.3389/fpsyg.2021.737882

(D) Reviewer offers the following advice; the authors should check the text carefully if the pieces of advice given here are appropriate. The paper needs complete rewriting.

(1) An Introduction should contain the following three parts:

Background: The author has to clarify the context. Ideally, the author should give an idea of the state-of-the-art of topic. In what context does this contribution enter? 

The Problem: If there were no problems, there would be no reason for writing a manuscript and no reason for reading it. So, please tell readers why they should proceed with reading.

The Proposed Solution: the author may outline the contribution of the manuscript. Here, the author has to make sure readers point out the novel's contributions and compare them with past results. The author should place the paper in proper context by citing relevant papers.

Results and discussion section: The presentation of results should be simple and straightforward in style. It would help if you improve your analysis and also present the comparison between the performance of your approach and other research. Results given in figures should not be repeated in tables. Remember to include a subsection or section ``Results and discussion section''. This section reports the most important findings, including statistical analyses as appropriate. It is very important to prove that your manuscript has a significant value and is not trivial. This paper seems to be written in a rush.

The figures are blurry: it seems that the authors are using raster graphics (poor quality) instead of vector graphics (best quality); see a discussion at https://en.wikipedia.org/wiki/Vector_graphics

Associate Editor recommends the authors study carefully the suggestions made in the paper ``Ten Simple Rules for Better Figures", see https://doi.org/10.1371/journal.pcbi.1003833 By doing so, the authors can learn how to design professional-quality figures.

Author Response

The authors use tools available in the literature (e.g., SieBERT) to classify sentiment in tweets.

Point 1: (A) I'm unsure about what is the main paper's novelty. Is it only about the classification of the three themes?

Response 1: Thank you for the helpful comment. To improve the clarity on this, we have added further explanations on the rationale for the present study in our introduction section. As highlight by an earlier study, which analysed temporal variations in the prevalence of the barriers and facilitators to COVID-19 vaccination over an 11-week study period, the barriers for individuals appeared to be consistent over time. Moreover, as COVID-19 vaccine acceptance rates are known to change over time, the current investigation would provide the most recent update on the current public concerns related to vaccine hesitancy, globally, which may inform policymakers on the subsequent approach to address prevailing public concerns on COVID-19 vaccination.

In terms of our study methods, we made use of BERT, which is currently the state-of-the-art technology in natural language processing (NLP), developed by Google in 2019 and now deployed in Google search engine for all English-language queries. This is in contrast to the traditional approach of bag-of-words in NLP, which does not take into account the context of the words within each sentence.

With regard to the novelty of the study findings, notably, a high percentage of the tweets identified in this study was related to the introduction of vaccine passports. Some countries have either introduced or are contemplating mandatory COVID-19 certification or passports. Although such policies could directly promote behaviour change and increase vaccine uptake, they may also create invidious comparisons (especially with implementation and equity issues) and produce strong negative sentiments as a result of perceived disparity and curtailment of individual freedom. This finding is in line with recent publications, which showed that vaccine mandates or its equivalents are powerful tools to bolster vaccination uptake, however, governments need to consider the strong negative reactions as a result of such policies, carefully weighing pre-existing COVID-19 vaccination coverage and hesitancy and the pandemic trajectory. There is also evidence that such coercive COVID-19 vaccine policies may further erode public trust and vaccine confidence and exacerbate socioeconomic inequalities as these policies may also alienate those who have yet to receive the vaccine and make them even less likely to do so.

Point 2: (B) Themes 2 and 3 are insignificant statistically compared to theme 1. It means that almost all posts are in Theme 1, and your classification tool is worthless because, in almost all cases, the correct topic is Theme 1.

Response 2: Thank you for the comment. Whilst it is true that the bulk of the posts identified were clustered semantically around Topic 1 and Theme 1, such analyses may not be as in-depth as manually conducted qualitative analyses. Moreover, qualitative analyses are less about numbers (n’s) and more about the ability of data to provide a rich and nuanced account of the phenomenon under study. As such, we do believe that Themes 2 and 3 are still very much relevant when it comes to gaining a better understanding of the prevailing negative sentiments towards COVID-19 vaccination.

Point 3: (C) Look at other studies about sentiment analysis related to Covid-19, and improve your review on the topic; an example is http://dx.doi.org/10.3389/fpsyg.2021.737882

Response 3: Thank you for the reference. We have read it, referenced it and taken pointers from the review.

Point 4: (D) Reviewer offers the following advice; the authors should check the text carefully if the pieces of advice given here are appropriate. The paper needs complete rewriting.

(1) An Introduction should contain the following three parts:

Background: The author has to clarify the context. Ideally, the author should give an idea of the state-of-the-art of topic. In what context does this contribution enter?

The Problem: If there were no problems, there would be no reason for writing a manuscript and no reason for reading it. So, please tell readers why they should proceed with reading.

The Proposed Solution: the author may outline the contribution of the manuscript. Here, the author has to make sure readers point out the novel's contributions and compare them with past results. The author should place the paper in proper context by citing relevant papers.

Response 4: Thank you for the comments. We have further expanded our introduction section and laid out the context and rationale for the present article.

Point 5: Results and discussion section: The presentation of results should be simple and straightforward in style. It would help if you improve your analysis and also present the comparison between the performance of your approach and other research. Results given in figures should not be repeated in tables. Remember to include a subsection or section ``Results and discussion section''. This section reports the most important findings, including statistical analyses as appropriate. It is very important to prove that your manuscript has a significant value and is not trivial. This paper seems to be written in a rush.

Response 5: Thank you for the comments, and apologize for the oversights. We have further expanded the discussion on how the topics fit into the Increasing Vaccination Model.

Point 6: The figures are blurry: it seems that the authors are using raster graphics (poor quality) instead of vector graphics (best quality); see a discussion at https://en.wikipedia.org/wiki/Vector_graphics

Associate Editor recommends the authors study carefully the suggestions made in the paper ``Ten Simple Rules for Better Figures", see https://doi.org/10.1371/journal.pcbi.1003833 By doing so, the authors can learn how to design professional-quality figures.

Response 6: Thank you for the comment and information. We have improved the image quality of all Figures in the paper.

Reviewer 2 Report

This paper aims to examine the key negative sentiments towards COVID-19 vaccination via the analysis of public Twitter posts over a year. Suggestions are provided as follows:

1. How should the period of study be considered? Does the choice of time period affect the results of the study? If so, what is the impact?

2. The introduction of the method is too simple, hope the author will further elaborate based on this article.

3. The main part is relatively thin, and it is recommended that the author further digs into the existing data for in-depth analysis after generating six related topics.

4. The logical relationship in Figure 2 proposed in the discussion section is relatively simple, and its promotion or inhibition relationship can be further explored in conjunction with the description of the topic in the discussion.

5. Interpretations of different topics in the discussion can also be derived without a thematic analysis. It is hoped that the author will revise and polish the article based on fully understanding the problems addressed by this study.

6. Possible problems with the methods used in this study as well as further solutions can also be detailed in the Discussion.

Author Response

Point 1: How should the period of study be considered? Does the choice of time period affect the results of the study? If so, what is the impact?

Response 1: This study period was originally from 1st April 2021 to 1st April 2022. We have now extended the study period to 1st August 2022. As COVID-19 vaccine acceptance rates are known to change over time, the current investigation would provide the most recent update on the current status of vaccine hesitancy, globally.

Point 2: The introduction of the method is too simple, hope the author will further elaborate based on this article.

Response 2: Thank you for the comment. We have further expanded our introduction section and laid out the context and rationale for the present article. “This study thus aimed to examine the key negative sentiments towards COVID-19 vaccination that are prevailing in the community via the analysis of public twitter posts over a one-year period (1 April 2021 to 1 August 2022). As COVID-19 vaccine acceptance rates are known to change over time [9], the current investigation would pro-vide the most recent update on the current status of vaccine hesitancy, globally. In doing so, policymakers can become more attuned to the public’s negative sentiments and work to address these concerns.”

Point 3: The main part is relatively thin, and it is recommended that the author further digs into the existing data for in-depth analysis after generating six related topics.

Response 3: Thank you for the comment. We have further explored and analysed the temporal trend for these themes over time, based on the number of tweets posted for each topic (Figure 3). Theme 1 consisted of the tweets posted under Topics 1 and 2, Theme 2 comprised of the tweets posted under Topics 3 and 4, and Theme 3 consisted of the tweets posted under Topics 5 and 6. These trends were relatively constant over the 16-month study period. Predictably, tweets pertaining to concerns for the effectiveness of the COVID-19 vaccines against emerging variants appeared to coincide with the surge in Delta (July to August 2021) and Omicron (December 2021) cases in the United States (US).

Point 4: The logical relationship in Figure 2 proposed in the discussion section is relatively simple, and its promotion or inhibition relationship can be further explored in conjunction with the description of the topic in the discussion.

Response 4: Thank you for the comment. We have further expanded the discussion on how the topics fit into the Increasing Vaccination Model.

Point 5: Interpretations of different topics in the discussion can also be derived without a thematic analysis. It is hoped that the author will revise and polish the article based on fully understanding the problems addressed by this study.

Response 5: Thank you for the comment. We have polished the article as per your comment.

Point 6: Possible problems with the methods used in this study as well as further solutions can also be detailed in the Discussion.

Response 6: Thank you for the comment. We have detailed the study shortcomings and limitations of the methods in our discussion section, “Nevertheless, the findings of the present study should be interpreted in light of the following limitations. First, the analysis was based solely on Twitter posts (with the majority of users hailing from North America and Europe) and only tweets in English were eligible for inclusion, hence the findings should not and cannot be generalized to all populations and communities. Moreover, public sentiments may also differ significantly depending on local socio-political factors and the pandemic situation at that time point. Second, the unsupervised machine learning approach and Python package were employed to process the social media data and identify tweets with ‘probably’ negative sentiments. This was based on probability and the unsupervised machine learning approach could not detect sarcasm, hence misclassification can occur as a result, albeit the methods have good supporting evidence for their utility and face validity [2, 27]. Third, nonhuman Twitter users (i.e., bots) masquerading as legitimate users and with the intention to distort public opinions [28] could have been included in our study sample. Despite best effort attempts to include tweets by users with actual human names and excluding retweets and duplicate tweets, a small number of these nonhuman tweets could have been included in our final study sample and unduly influenced the topic modelling process.”

Reviewer 3 Report

This paper attempts to examine the key negative sentiments towards COVID-19 vaccination via the analysis of public twitter posts over a one-year period. State-of-the-art approaches for text mining/natural language processing are employed. After reading the paper, the reviewer generally supports the publication of this work and recommends that the authors improve the paper presentation according to the following points:

1) Please state the region or area of the survey (e.g. only in Singapore or the whole English speaking countries).

2) Please extend the current literature review section regarding recent applications of text mining/natural language processing methods in examining public sentiments.

3) Please review recent applications of text mining/natural language processing methods in examining public sentiments towards COVID-19 vaccination.

4) What are the existing works? What are the used methods? What are their limitations? Please elaborate these points.

5) In section 2: provide a paragraph to discuss the use of the Bidirectional Encoder Representations from Transformers (BERT) in the proposed system.

6) In section 2: provide a paragraph to discuss the use of the SieBERT in the proposed system.

7) In section 2: It is beneficial for the readers of the journal that a figure summarizing the proposed system is provided.

8) In section 3: provide more details regarding the components of the flowchart, e.g. how to identify duplicate tweets? …

Author Response

Point 1: Please state the region or area of the survey (e.g. only in Singapore or the whole English speaking countries).

Response 1:  We apologize for this oversight, and has since added the following sentence in the Methods section to make clear to readers on the country of origin, “there was no restriction in the country of origin of the tweets, as long as the tweets were posted in English language.”

Point 2: Please extend the current literature review section regarding recent applications of text mining/natural language processing methods in examining public sentiments.

Response 2: Thank you for the comment. We have overviewed recent applications of similar methods in examining public sentiments, and added the following section (along with 3 additional references 2, 7, 8): “Previous studies have found such social media analyses to be a feasible and novel method to study public sentiment and emotional manifestations on a given topic [2,7,8]”.

Point 3: Please review recent applications of text mining/natural language processing methods in examining public sentiments towards COVID-19 vaccination.

Response 3: Thank you for the comment. We have overviewed recent applications of similar methods in examining public sentiments towards COVID-19 vaccination, “previous studies have found rampant and persistently negative sentiments and misinformation on Twitter, which may adversely influence individual views and result in vaccine hesitancy or refusal [2,4,5]” and “As highlight by an earlier study [2], which analysed temporal variations in the prevalence of the barriers and facilitators to COVID-19 vaccination over a 11-week study period, the barriers for individuals appeared to be consistent over time”.

Point 4: What are the existing works? What are the used methods? What are their limitations? Please elaborate these points.

Thank you for the comment. As recommended, we have detailed the existing works, used methods and limitations of the methods in our discussion section, “Nevertheless, the findings of the present study should be interpreted in light of the following limitations. First, the analysis was based solely on Twitter posts (with majority of users hailing from North America and Europe) and only tweets in English were eligible for inclusion, hence the findings should not and cannot be generalized to all populations and communities. Moreover, public sentiments may also differ significantly depending on local socio-political factors and the pandemic situation at that time-point. Second, the unsupervised machine learning approach and Python package were employed to process the social media data and identify tweets with ‘probably’ negative sentiments using a rule-based approach. This was based on probability and the unsupervised machine learning approach could not detect sarcasm, hence misclassification can occur as a result, albeit the methods have good supporting evidence for their utility and face validity [2, 27]. Third, nonhuman Twitter users (i.e., bots) masquerading as legitimate users and with the intention to distort public opinions [28] could have been included in our study sample. Despite best effort attempts to include tweets by users with actual human names and excluding retweets and duplicate tweets, a small number of these nonhuman tweets could have been included in our final study sample and unduly influenced the topic modelling process.”

Point 5: In section 2: provide a paragraph to discuss the use of the Bidirectional Encoder Representations from Transformers (BERT) in the proposed system.

Response 5: Thank you for the comment. We have further discussed the use of BERT in our methods section: “Bidirectional Encoder Representations from Transformers (BERT), a state-of-the-art deep machine learning approach for natural language processing (NLP), uses unsu-pervised masked language model (MLM) and the unsupervised next sentence predic-tion (NSP) for text deep pre-training and fine-tuning [10] compared to the traditional bag-of-words model in NLP. For MLM, a percentage of words are masked at random and then these masked words are predicted based on their bidirectional context; whereas the NSP model pre-trains the text to understand relationships between sen-tences [10]. Unlike other text analysis approaches that require text to be pre-processed as part of data preparation prior to the analysis, the BERT-based model does not require much text pre-processing as the sentence context is provided by BERT.”

Point 6: In section 2: provide a paragraph to discuss the use of the SieBERT in the proposed system.

Response 6: Thank you for the comment. We have further discussed the use of SieBERT in our methods section: “The SieBERT, a pre-trained sentiment in English analysis model, was performed and only negative sentiments tweets were selected [12].”

Point 7: In section 2: It is beneficial for the readers of the journal that a figure summarizing the proposed system is provided.

Response 7: Thank you for the comment. As recommended, we have illustrated the methods used in Figure 1.

Point 8: In section 3: provide more details regarding the components of the flowchart, e.g. how to identify duplicate tweets? …

Response 8: Tweets that are identical in the sentence/words were identified as duplicate tweets and excluded.

Reviewer 4 Report

This work examined the key negative sentiments towards COVID-19 vaccination via analyzing 16 public Twitter posts over one year. Original tweets (in English) were extracted from 1st April 2021 to 1st April 17, 2022. A Bidirectional Encoder Representations from Transformers (BERT)-based 18 model was applied, and only negative sentiments tweets were selected. The paper's contribution to existing knowledge in this research field is well justified. The paper does not contribute enough, and the following points can improve the manuscript.

1.     Abstract and introduction should be enhanced to show the importance of the work.

2.     Enhance the English of the work. There are too many problems with paper typesetting.

3.     Add related work section and add comparison in table format in the related work section.

4.     Table 1 should be updated, and most of the text should be shifted to the text.

5.     The applied Methods should be discussed mathematically to show the readers how to use them.

6.     The output results and conclusions should be compared with other methods in the literature.

7.     The manuscript organization should be improved. 

8.     Change the “Conclusion” section title to “conclusion and future directions” and add more future directions to the research.

9.     The paper is not acceptable in its current form. The article needs rewriting to address the comments mentioned above. 

Author Response

Point 1: Abstract and introduction should be enhanced to show the importance of the work.

Response 1: Thank you for the comment. We have further expanded our introduction section and laid out the context and rationale for the present article.

Point 2: Enhance the English of the work. There are too many problems with paper typesetting.

Response 2: Thank you for the comment. We have done a close read and edit of the entire manuscript for typesetting, language and grammar.

Point 3: Add related work section and add comparison in table format in the related work section.

Response 3: Thank you for the comment. We have overviewed recent applications of similar methods in examining public sentiments towards COVID-19 vaccination, “previous studies have found rampant and persistently negative sentiments and misinformation on Twitter, which may adversely influence individual views and result in vaccine hesitancy or refusal [2,4,5]” and “As highlight by an earlier study [2], which analysed temporal variations in the prevalence of the barriers and facilitators to COVID-19 vaccination over an 11-week study period, the barriers for individuals appeared to be consistent over time”.

Point 4: Table 1 should be updated, and most of the text should be shifted to the text.

Response 4: Thank you for the comment. We have revised Table 1 as per your comments.

Point 5: The applied Methods should be discussed mathematically to show the readers how to use them.

Response 5: Thank you for the comment. We have further discussed our methods and illustrated them in Figure 1 as well.

Point 6: The output results and conclusions should be compared with other methods in the literature.

Response 6: Thank you for the comment. We have acknowledged the limitations of our methods in the discussion section. It is also that other newer techniques of sentiment analysis, especially those based on supervised deep learning, may achieve better accuracy in sentiment analysis. However, the development of new models for sentiment analysis is a separate area, beyond the scope of the present study.

Point 7: The manuscript organization should be improved.

Response 7: Thank you for the comment. As recommended, we have improved the organisation of the manuscript as per the journal’s guidelines.

Point 8: Change the “Conclusion” section title to “conclusion and future directions” and add more future directions to the research.

Response 8: Thank you for the comment. We have amended the section title as per your recommendation and added more future directions to the research, “For future work, topic-based sentiment analysis of texts from other social media platforms including Facebook and Reddit could be studied as well. Other methods could be compared as well as it is possible that newer techniques of sentiment analysis may yield greater accuracy.”

Point 9: The paper is not acceptable in its current form. The article needs rewriting to address the comments mentioned above.

Response 9: We are grateful to the reviewer for the many constructive recommendations to improve the manuscript, and apologize for oversights. We hope we have addressed your comments mentioned above, and will be glad to improve further too following any additional recommendations.

Round 2

Reviewer 1 Report

None

Author Response

Thank you.

Reviewer 4 Report

The authors have addressed most of my concerns. The paper can be accepted for publication.

Author Response

Thank you.